# The Intended Uses of Automated Fact-Checking Artefacts:
# Why, How and Who

## Michael Schlichtkrull[1], Nedjma Ousidhoum[1,2], Andreas Vlachos[1]

[1]Department of Computer Science and Technology, University of Cambridge, [2]Cardiff University
mss84@cam.ac.uk  OusidhoumN@cardiff.ac.uk  av308@cam.ac.uk

## Abstract

Automated fact-checking is often presented as an epistemic tool that fact-checkers, social media consumers, and other stakeholders can use to fight misinformation. Nevertheless, few papers thoroughly discuss *how*. We document this by analysing 100 highly-cited papers, and annotating epistemic elements related to intended use, i.e., means, ends, and stakeholders. We find that narratives leaving out some of these aspects are common, that many papers propose inconsistent means and ends, and that the feasibility of suggested strategies rarely has empirical backing. We argue that this vagueness actively hinders the technology from reaching its goals, as it encourages overclaiming, limits criticism, and prevents stakeholder feedback. Accordingly, we provide several recommendations for thinking and writing about the use of fact-checking artefacts.

## 1 Introduction

Following an increased public interest in online misinformation and ways to fight it, fact-checking has become indispensable (Arnold, 2020). This has been matched by a corresponding surge in NLP work tackling *automated fact-checking* and related tasks, such as rumour detection and deception detection (Guo et al., 2022). In the process, many models, datasets, and applications have been created, referred to as *artefacts*. A key part of technological artefacts are their intended uses (Hilpinen, 2008; Kroes and van de Poel, 2014). Some have argued that artefacts can *only* be fully understood through these (Krippendorff, 1989). Spinoza (1660) argued that a hatchet which does not work for its intended purpose – chopping wood – is no longer a hatchet. Heidegger (1927) went further, arguing that artefacts can only be properly *understood* when actively used for their intended purpose; i.e., the only way to understand a hammer is to hammer with it.

Figure 1: Annotated quotes from Wang (2017). We highlight the motivation of the work in blue, and the model means (classification) in yellow. The goal stated (epistemic ends) is to limit misinformation. However, the authors do not state the data actors (*who*) and the application means (*how*) for reaching this goal.

Automated fact-checking artefacts are no different. The majority of papers envision them as epistemic tools to limit misinformation. In our analysis, we find that 82 out of 100 automated fact-checking papers are motivated as such. Unfortunately, many papers only discuss how this will be achieved in vague terms – authors argue *that* automated fact-checking will be used against misinformation, but not *how* or by *whom* (see Figure 1).

Connecting research to potential use allows researchers to shape their work in ways that take into account the expressed needs of key stakeholders, such as professional fact-checkers (Nakov et al., 2021). It also enables critical work (Haraway, 1988), and facilitates thinking about unintended shortcomings, e.g., dual use (Leins et al., 2020;

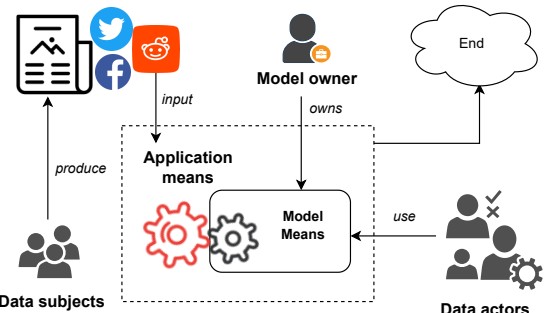

Figure 2: Diagram of epistemic elements in automated fact-checking narratives. For example, "*journalists* (data actors) should use a *classification model* (modeling means) owned by *a media company* (model owner) to *triage claims* (application means) made by *politicians* (data subjects), in order to *limit misinformation* (ends)." This constitutes a *narrative*, the efficacy of which may have *feasibility support* (e.g., a user study).

Kaffee et al., 2023). Grodzinsky et al. (2012) argue that *"people who knowingly design, develop, deploy, or use a computing artifact can do so responsibly only when they make a reasonable effort to take into account the sociotechnical systems in which the artifact is embedded"*. Underspecification risks overclaiming, as authors cannot estimate if the aim they *claim* to work towards can actually be met by the means they employ.

For automated fact-checking this is especially problematic. Overclaiming can create misinformation about misinformation, resulting in the opposite effect from the one intended (Altay et al., 2023). We put forward the notion that the development of automated fact-checking into a useful tool for fighting misinformation will be more constructive with a clearer vision of intended uses.

In this paper, we investigate narratives about intended use in automated fact-checking papers. We focus on those where fact-checking artefacts are introduced as epistemic tools, i.e., where the aim is to increase the quality or quantity of knowledge possessed by people. We analyse 100 highly cited papers on the topic. We manually annotate the epistemic elements of their narratives (shown in Figure 2), relying on content analysis (Bengtsson, 2016; Krippendorff, 2018). Specifically, we extract the stakeholders whose epistemic experience will be affected (i.e., *data subjects* and *data actors*), the strategies the authors propose (i.e., *application* and *model means*), and their intended goals (i.e., *ends*). We categorise the narratives extracted by analysing the links between these elements.

We find that many papers contain narratives with similar types of vagueness, leaving out key details such as the actors involved. Based on this, we give recommendations to clarify discussions of intended use: 1) clearly state data actors, 2) clearly state application means, 3) account for stakeholders beyond journalists, 4) ensure that the epistemic narratives are coherent and supported by relevant literature, and 5) empirically study the effectiveness of the proposed uses.

## 2 Related Work

A series of recent studies investigate the values and practices of NLP and machine learning research. Birhane (2021) argues that current practices in machine learning often follow historically and socially held norms, conventions, and stereotypes when they make predictions. Birhane et al. (2022) further study the values encoded in ML research by annotating influential papers, finding that, while research is often framed as neutral, resulting artefacts can deeply affect institutions and communities.

Several papers have recently examined assistive uses of NLP. Buçinca et al. (2021) found that users frequently "overrely" on AI/ML tools, accepting suggestions even when they are wrong without double-checking. Consequently, Perry et al. (2022) found that coding assistants can lead to insecure code. However, Vasconcelos et al. (2023) showed that explanations could reduce this effect – albeit only so long as the explanations are designed to be easily understandable and verifiable by the user. Similar to our criticism of research in automated fact-checking, Gooding (2022) argued that, for text simplification, the disconnect between algorithms and their applications hindered discussions of ethics.

In automated fact-checking, Nakov et al. (2021); Konstantinovskiy et al. (2021) have recently attempted to connect research to users by examining the concerns of professional fact-checkers. Generally, they find that organisations desire automation beyond veracity prediction models, which cannot function in isolation of media monitoring, claim detection, and evidence retrieval. Along parallel lines, Glockner et al. (2022) found that many automated fact-checking datasets make unrealistic assumptions about what information is available at inference time, limiting the usefulness of the resulting artefacts (i.e., systems and datasets) in real-world scenarios.

Search engines have previously been studied as epistemic tools, with findings generalising to automated fact-checking. Simpson (2013) argued that search engines perform the roles of *surrogate experts*, connecting information seekers to testimony, and choosing how to prioritise testimonial authorities. Miller and Record (2013, 2017) further argued that search engine providers, developers, and researchers therefore hold responsibility for beliefs formed by users, as the internal workings of systems are not accessible to users. There are documented real-world consequences from this opacity, e.g., Machill and Beiler (2009) found evidence of bias in journalistic publications due to overreliance on biased ranking algorithms.

## 3   Content Analysis

To understand the narratives of the fact-checking literature, we perform a content analysis following Bengtsson (2016); Birhane et al. (2022) by means of annotating 100 highly-cited papers. As recommended by Krippendorff (2018), we annotated the data in two rounds, working *inductively* in the first round. That is, during the first pass we began with a small set of labels for each element, and added new as necessary. We then unified the two sets of labels (i.e., the old and new sets of labels), merging labels where necessary. For the second pass, we re-annotated each paper with labels from the unified list, adding no new labels.

### 3.1   Annotation Scheme

We start our analysis by extracting *quotes* from the introductory sections of papers. These are spans starting and ending at sentence boundaries (although potentially spanning multiple sentences within a paragraph). We identify for further processing those that either discuss *what* a paper does or *why* the authors focus on it. Then, we annotate them with epistemic elements. This annotation is multi-label, i.e., for each element, one quote may have several (or no) labels. The elements (in **bold**) can be seen here along with an example of a label (in *italic*):

- **data subjects**: the people whose behaviours are analysed (e.g., *social media users*),
- **data actors**: the people who are intended to use the model outputs (e.g., *journalists*),
- **model owners**: people/organisations who own the model and may make choices about its deployment (e.g. *social media companies*),

- **modeling means**: what machine learning approaches the paper takes (e.g., *classification*),
- **application means**: how the authors want to apply the model to accomplish the goal (e.g., by *triaging claims*),
- **ends**: the purpose of the designed system (e.g., *limiting misinformation*).

We add a further level of annotation at the discourse-level – that is, spanning and including epistemic elements from multiple quotes. Here, we extract **epistemic narratives**, i.e., the stories about knowledge told in each paper (e.g., *automated content moderation*). Each narrative is associated with a specific set of data actors (e.g., journalists). Narratives can be associated with multiple elements, and elements can be part of multiple narratives. We also extract the **feasibility support** for each narrative, i.e., any backing for the feasibility or efficacy of that narrative (e.g., *scientific research*). See the illustration in Figure 2 for an overview of our framework.

### 3.2   Annotation Process

The annotation was conducted by the first two authors, NLP experts working on automated fact-checking. After the first annotation round, we clarified the definitions of each category, designed a flowchart to improve the consistency of the narrative-level annotations (see Appendix B.5), and re-annotated the data. This process yielded a substantial Krippendorff-$\alpha$ agreement score of 0.76 on our narrative annotations, using Jaccard distance as the metric. We provide the list of papers and full set of annotations at https://github.com/MichSchli/IntendedAFCUses, and the full annotation guidelines in Appendix B.

We collected the papers for annotation using the continually updated repository of papers[1] focusing on fact-checking and related tasks (e.g., rumor detection, deception detection), which accompanies the recent survey by Guo et al. (2022). To focus our investigation on influential narratives, we limited our analysis to the 100 most cited papers listed in the repository. We manually extracted quotes from the introduction sections of each paper, and when important elements were missing, we additionally considered quotes from the abstract.

---

[1]https://github.com/Cartus/Automated-Fact-Checking-Resources

## 4 Extracted Epistemic Elements

We extracted the following lists of labels for epistemic elements, along with the percentage of narratives they appeared in. Note that the annotation is multilabel, so the percentages do not sum to 1. Definitions of each label can be found in Appendix B.2.

**Data Subjects** *Social media users (24.9%), professional journalists (15%), politicians & public figures (5.6%), product reviewers (1.7%), technical writers (1.7%), citizen journalists (0.4%).*

**Data Actors** *Professional journalists (18.5%), citizen journalists (2.6%), media consumers (4.7%), scientists (3.9%), algorithms (1.7%), engineers & curators (1.3%), law enforcement (0.9%).*

**Model Owners** *Social media companies (4.7%), scientists (1.3%), law enforcement (0.9%), traditional media companies (0.4%).*

**Modeling means** *Classify/score veracity (79.3%), evidence retrieval (24.5%), produce justifications (6.9%), human in the loop (11.2%), classify/score stance (3.9%), corpora analysis (2.1%), generate claims (0.9%).*

**Application means** *Identify claims (26.2%), provide veracity scores (17.5%), supplant human fact-checkers (13.7%), gather and present evidence (10.7%), present aggregates of social media comments (6.9%), triage claims (6.0%), vague persuasion (3.4%), filter system outputs (3.0%), analyse data (3.0%), automated removal (1.7%), identify multimodal inconsistencies (1.7%), maintain consistency with a knowledge base (1.3%), produce misinformation (0.9%).*

**Ends** *Limit misinformation (78.1%), increase veracity of published content (7.2%), limit AI-generated misinformation (3.4%), develop knowledge of NLP/language (3.4%), detect falsehood for law enforcement (0.9%), avoid biases of human fact-checkers (0.4%).*

## 5 Extracted Narratives

At the discourse level, we extracted *epistemic narratives*. Below, we give a short definition of each. We also calculated the percentage of papers each narrative appears in – see Figure 4. Note that this annotation is also multilabel, so the percentages do not sum to 1.

> *"The dissemination of fake news may cause large-scale negative effects, and sometimes can affect or even manipulate important public events. [...] Therefore, it is in great need of an automatic detector to mitigate the serious negative effects caused by the fake news."*
> – **vague identification** in Wang et al. (2018).
>
> *"The ever-increasing amounts of textual information available combined with the ease in sharing it through the web has increased the demand for verification, also referred to as fact checking. [...] In this paper, we introduce a new dataset..."*
> – **vague opposition** in Thorne et al. (2018).
>
> *"Rumours are rife on the web. False claims affect people's perceptions of events and their behaviour, sometimes in harmful ways [...] While breaking news unfold, gathering opinions and evidence from as many sources as possible as communities react becomes crucial to determine the veracity of rumours and consequently reduce the impact of the spread of misinformation."*
> – **vague debunking** in Derczynski et al. (2017).

Figure 3: Examples of vague narratives in highly cited automated fact-checking papers.

### 5.1 Vague narratives

We find that the majority of papers, 56%, contextualize their research through narratives we characterize as *vague*, which make it difficult to reason about or criticize the potential applications of the proposed technologies. We identify three primary "types" of vagueness, discussed below. See Figure 3 for an example of each.

**Vague identification (31%)** The most common form of vagueness we identify is a narrative where automated fact-checking will be used to *identify* potential misinformation. However, no discussion is made of what will happen to potential misinformation after flagging it. As many options are available – automated removal, warning labels, additional work by professional journalists – it is difficult to assess the impact of these applications.

**Vague opposition (19%)** Automated fact-checking is presented as a tool to fight misinformation with no discussion of how that tool will actually be used. Crucially, no *application means* are mentioned, and other than the final goal – opposition to misinformation – the intended uses of the artefacts introduced are not discussed.

**Vague debunking (14%)** Authors clearly intend their artefacts to aid in the production of evidence-based refutations, i.e., *debunking*, similar to the

function of professional fact-checkers. However, it is not clear whether the system is intended to replace human fact-checkers, or to assist them. No *data actors* are mentioned. The application means are typically to present evidence or veracity scores to users (45% and 40% of cases, respectively).

## 5.2 Clear Narratives

In addition to the vague narratives, we furthermore identify several clearly stated intended uses. We also calculated the percentage of papers each narrative appears in. As previously stated, a paper can have many narratives, so the percentages do not sum to 1.

**Automated external fact-checking (22%)** Artefacts developed in the paper are intended to *fully* automate the fact-checking process, without human-in-the-loop interventions.

**Assisted external fact-checking (18%)** Artefacts are intended as assistive tools for professional or citizen journalists, deployed for *external* fact-checking (i.e., assistance in writing fact-checking articles).

**Assisted media consumption (8%)** Artefacts are intended as assistive tools for *consumers* of information, either as a layer adding extra information to other media or as a standalone site where claims can be tested.

**Scientific curiosity (8%)** Artefacts in the paper are not intended for use. Instead, the process of development itself is claimed to produce knowledge i.e., about language or misinformation.

**Assisted knowledge curation (7%)** Artefacts are intended as assistive tools for the curation of large knowledge stores, such as Wikipedia.

**Assisted internal fact-checking (4%)** Artefacts are intended as an assistive tool for professional or citizen journalists, deployed for *internal* fact-checking (i.e., assistance in fact-checking articles before publication to ensure quality).

**Automated content moderation (4%)** Artefacts are intended to complement or replace human moderators on e.g., social media sites, by performing moderation functions autonomously.

**Truth-telling for law enforcement (1%)** Artefacts are intended for use by law-enforcement groups or in courtrooms as lie detectors.

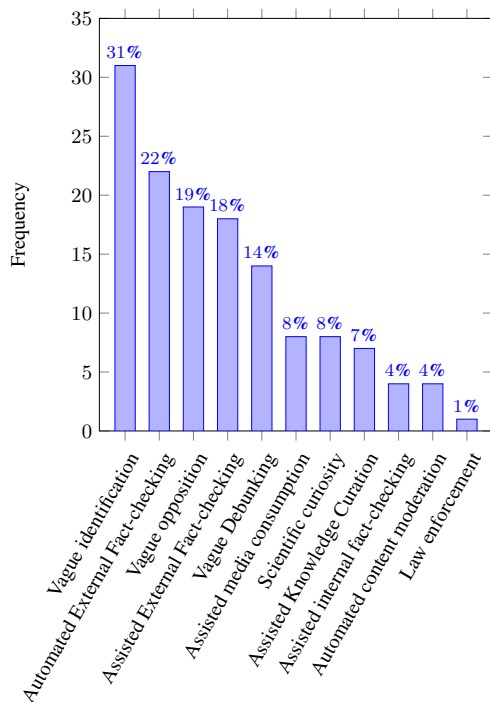

Figure 4: Frequency of each epistemic narrative found in our analysis of 100 automated fact-checking papers.

## 6 Findings

Based on our content analysis, we identify several areas of concern in the analysed papers. In the following sections, we discuss our main findings.

### 6.1 Stakeholders

Technological artefacts are components in sociotechnical systems involving agents (van de Poel, 2020). A crucial consideration is therefore the *stakeholders* who will be involved in deployment: data subjects, data actors, and model owners.

**Who are the fact-checking artefacts for?** A key factor of *vague* narratives is the omission of data actors. For clear narratives, we find mentions of data actors in 51% of cases. For vague narratives, this number is dramatically lower: 6.7% for vague opposition, 7.1% for vague identification, and 0% for vague debunking. Model owners are discussed in 10.2% of non-vague narratives, and 2.8% of vague narratives. Intuitively, this is reasonable: thinking about *who* will use a technology also encourages authors to think about *how* it will be used. Missing data actors is especially problematic for work that seeks to provide explainability, where explanations can be understood differently by different users (Schuff et al., 2022). We find that only a small minority, 6.25%, of narratives that include

justification production include any discussion of data actors (professional journalists, in all cases).

**And who are we fact-checking, anyway?** A concern in AI ethics is the lack of attention paid to *data subjects*, i.e., the people whose data is input to AI systems (Kalluri, 2020). We find that data subjects are discussed in 45% of non-vague and 39% of vague narratives. However, actors and subjects are rarely the same entities – there is overlap only in 8% of non-vague narratives, and no vague narratives. Further, actors tend to be experts (professional journalists, in 62% of all narratives), rather than peers of the data subjects (social media users, in 65% of narratives). This mirrors Kalluri (2020), who argues that *"researchers in AI overwhelmingly focus on providing highly accurate information to decision makers. Remarkably little research focuses on serving data subjects."*

**What about other uses?** Combating misinformation is by far the most common end. One narrative has a slightly different focus, *assisted knowledge curation*, where the aim in 84% of cases is to prevent errors in published content. 7% of papers envision applications to curation, including for relational knowledge bases as well as for Wikipedia. The typical application means is to filter the output of *other* systems, e.g., knowledge base completion models. Survey papers, e.g., Thorne and Vlachos (2018); Lazer et al. (2018); Kotonya and Toni (2020); Hardalov et al. (2022); Guo et al. (2022) tend not to mention this potential use, even though they go into detail with others.

## 6.2 Mismatched means

Even in clear narratives, there is often a disconnect between the proposed applications and the actual NLP techniques used. For example, 81% of papers seeking to supplant professional fact-checkers, i.e., replicate their entire function, rely on classification, and only 31% include evidence retrieval. High-quality fact-checks primarily focus on finding and explaining available evidence, rather than on veracity labels (Amazeen, 2015; Konstantinovskiy et al., 2021). Professional fact-checking includes significant editorial work, and fact-checkers are expected to "never parrot or trust sources" (Silverman, 2013). Instead, fact-checkers reason about the provenance of their evidence, and convey that to their readers.

## 6.3 Unsupported ends

We frequently find a disconnect between application means and envisioned ends. For example, 50% of narratives of automated content moderation suggest direct, algorithmic removal of items classified as false. This is of questionable effectiveness. Removal can induce an increase in the spread of the same message on other platforms, or increase toxicity (Sanderson et al., 2021; Ribeiro et al., 2021). Censorship can also amplify misinformation via the "Streisand effect", where the attempt to hide information encourages people to seek it out (Jansen, 2015; The Royal Society, 2022). Moreover, algorithmic bias (Noble, 2018; Weidinger et al., 2022) or misuse by model owners (Teo, 2021) could further distort what is removed. Given that, we find it likely that automated removal would fail either by *not* reducing belief in misinformation, or by reducing belief in other *true* information. This is not isolated to a single narrative. For assisted external fact-checking, several papers suggest providing veracity labels or scores to human fact-checkers, who have expressed strong skepticism at the usefulness of such scores (Nakov et al., 2021). We provide further analysis in Appendix A.

## 6.4 Lack of feasibility support

Papers rarely provide evidence in support of the efficacy of their narratives – regardless of whether, as with, e.g., *assisted media consumption*, there is significant work on measured effectiveness (Clayton et al., 2020; Pennycook et al., 2020; Sanderson et al., 2021). Only 18% of *non-vague* narratives, and 0% of vague narratives rely on citations to prior work or relevant literature from other fields for this. Furthermore, when narratives *are* supported by citations, some references do not fully support the claims made. For example, Cohen et al. (2011) is cited to support the effectiveness of fully automated fact-checking, but all the original paper does is *suggest* that such a task might be relevant. That is, there is no underlying quantitative or qualitative data to determine effectiveness.

## 6.5 Evidence is not a silver bullet

The idea that humans should update their beliefs based on model predictions is common to several narratives. This can be problematic, as model predictions can be inaccurate – especially for machine-generated claims that can easily mimic "truthy" styles (Schuster et al., 2020). The commonly pro-

posed remedy is to rely on support from retrieved evidence (Thorne et al., 2018; Guo et al., 2022). This view of evidence as a silver bullet is epistemologically naive, and high retrieval performance is not enough to provide support for the end of *limiting misinformation*. We identify at least three ways in which retrieving and presenting even credible evidence can still produce epistemic harm.

First, the predictions made by retrieval systems can be self-fulfilling (Birhane, 2021). A system asserting that a particular piece of evidence is more credible or relevant than an alternative can make users agree with that choice, even if they would otherwise not have (Buçinca et al., 2021). This is an existing concern in journalism, known as "Google-ization": overreliance on search algorithms causes a "distortion of reality" when deadlines are tight (Machill and Beiler, 2009).

Second, if a system presents many retrieved documents without accounting for provenance, users may rely on sheer quantity to make quick (but possibly wrong) judgments about veracity. This is a common bias known as the *availability heuristic* (Schwarz and Jalbert, 2021). A retriever may have collected many documents from one source, or from an epistemic community structured around a "royal family of knowers" from whom many seemingly independent sources derive their information (Bala and Goyal, 1998; Zollman, 2007).

Finally, evidence can mislead if users hold prejudices that lead them to make wrongful inferences. Fricker (2007) cites *To Kill a Mockingbird* as an example: because of racial prejudice, evidence that a black man is *running* is interpreted as evidence that he is *guilty*. Conceivably, a poorly trained or heavily biased fact-checking system could similarly cause harm by drawing undue attention to irrelevant – but stereotyped – evidence.

Unfortunately, these harms tend to affect the already marginalised most (Fricker, 2007; O'Neil, 2016; Noble, 2018). As argued by Birhane (2021), *"current practices in machine learning often impose order on data by following historically and socially held norms, conventions, and stereotypes"*.

## 6.6 Explaining vagueness

It is difficult to speculate on why automated fact-checking papers are prone to vagueness. A proper answer would require interviewing the authors of the papers included in our analysis, i.e., to conduct ethnographic studies with researchers as in

Knorr Cetina (1999). We would guess, however, that some mix of the following factors could be potential reasons:

- Common datasets often either use synthetic, purpose-made claims generated e.g., from Wikipedia, rather than "real" misinformation (Thorne et al., 2018), or they do not provide annotations for evidence. Making clear, direct claims about efficacy on real misinformation is therefore difficult.
- Leaderboarding is easier than including data actors in evaluation (Rogers and Augenstein, 2020), and excluding actors may therefore be the path of least resistance. As an additional result, some authors may see the introduction and the motivation as less important sections than e.g., experimental results, and put less effort into their writing.
- To highly invested researchers, the effectiveness of their preferred means may seem obvious (Koslowski, 2012). I.e., they may think that – *"it is self-evident that automatically identifying potential misinformation will help us fight it"*.
- Engaging with the literature in psychology and journalism on the efficacy of fact-checking may be daunting for NLP researchers. Some of this literature may be not be accessible to researchers in other fields, who are nonetheless interested in helping via the automation of the process (Hayes, 1992; Plavén-Sigray et al., 2017).

## 7 Recommendations

Authors of fact-checking papers clearly believe their research to be solving a societal need: 82% of narratives in our analysis had "limiting misinformation" as the desired end, and an additional 7% had "increasing the veracity of published content". As we have argued, vagueness around intended use in current papers may prevent the community from contributing to these goals, as it can block input from key stakeholders or obscure discussion of risks. Our recommendations below focus on bringing research on automated fact-checking into closer alignment with its stated goal. We emphasise that our intention is not to create another checklist for NLP conferences, as clarity surrounding intended use is already covered in the recently adopted ARR

Responsible NLP Checklist[2] (Rogers et al., 2021). However, authors may find our recommendations useful for *meeting* that requirement.

## 7.1 Discuss data actors

A key component of vague narratives is the absence of data actors. Thinking about *who* will use a technology encourages thinking about *how* it will be used. As such, our first recommendation is to include a clear discussion of the intended users. This also enables input from relevant communities: researchers working on systems designed for professional fact-checkers can seek feedback from those fact-checkers, which is crucial for evaluation. A fact-checking model paired with a professional journalist constitutes a very different sociotechnical system from one paired with, for example, a social media moderator (van de Poel, 2020). If the aim is to evaluate the capacity for technologies to accomplish real-world aims, it is necessary to evaluate them *for specific users*.

We highlight this especially for systems that produce justifications or explanations, where recent work shows that biases and varying technical competence influences what explainees take from model explanations (Schuff et al., 2022). Justifications may be a powerful tool to counteract overreliance on, e.g., warning labels (Pennycook et al., 2020) or search rankings (Machill and Beiler, 2009), but may only be helpful if designed for relevant data actors (Vasconcelos et al., 2023).

We note that avoiding discussion of data actors (or other epistemic elements) can be seen as a variant of Haraway's (1988) "god trick", where authors avoid situating their research artefacts within the context of real-world uses. This prevents criticism of and accountability for any harm caused by applications of the technology. As actors hold a major part of the responsibility for the consequences of deploying the technology (Miller and Record, 2017), explicitly including them is necessary to discuss the ethics of use.

## 7.2 Discuss application means

Beyond the absence of *who* is intended to use automated fact-checking artefacts, some papers also lack a discussion of *how* they will be used. That is, the narratives do not have clear application means. This type of vagueness is particular to the

*vague opposition* narrative. Leaving out application means disconnects the fact-checking artefacts *entirely* from any real-world context. Taking Krippendorff's (1989) view that technological artefacts *must* be understood through their uses, this prevents any full understanding of these artefacts. We recommend including a clear discussion of application means in any justificatory narrative.

## 7.3 Account for different stakeholders

Where data actors are mentioned, they are mostly professional journalists (62%). They rarely overlap with the data subjects, except in the case of internal fact-checking. We note that professional journalists are predominantly white, male, and from wealthier backgrounds (Spilsbury, 2017; Lanosga et al., 2017). Misinformation often (1) targets and undermines marginalized groups or (2) results from inequality-driven mistrust among the historically marginalised (Jaiswal et al., 2020). Professional journalists may, as such, not be the only actors well-suited for fighting it.

In the spirit of developing systems for all potential stakeholders, we encourage authors to widen their conception of who systems can be designed for. Media consumers feature prominently as data subjects, but are rarely expected to act on model outputs. This may be a limiting factor for what researchers can accomplish in terms of opposing misinformation. The literature on countermessaging highlights the need for ordinary social media users to participate in interventions (Lewandowsky et al., 2020). Tripathy et al. (2010) suggest a model where ordinary users intervene against rumours by producing anti-rumours. They argue that this would be superior to traditional strategies, as it limits reliance on credibility signals from distant authorities (Druckman, 2001; Hartman and Weber, 2009; Berinsky, 2017).

Early work on automated fact-checking (Vlachos and Riedel, 2014) proposed the technology as a tool to enable ordinary citizens to do fact-checking – i.e., citizen journalism. Decentralised, community-driven fact-checking has been shown to represent a viable, scalable alternative to professional fact-checking (Saeed et al., 2022; Righes et al., 2023). Citizen journalists have unfortunately largely disappeared since then, appearing as actors only in three narratives (10%) of assisted external fact-checking and one narrative (11%) of assisted internal fact-checking. Similarly, human content

moderators are entirely missing. Data curators, although somewhat frequent in research, are often overlooked in survey papers. We suggest that studying the user needs of groups beyond journalists and social media companies is a fruitful area for future research. Further, it may be relevant to study who *spreads* misinformation (Mu and Aletras, 2020).

### 7.4 Ensure Narrative Coherence

In our analysis, we identified two unfortunate trends: inconsistencies between modeling and application means (see Section 6.2), and inconsistencies between means and ends (see Section 6.3). These are problematic, as they may in effect be overclaims, e.g., authors claim that a classifier will tackle a task that is not a classification task. In Section 6.4, we found that only a minority of papers draw on the literature to support the feasibility of their proposed narratives. There is significant literature on interventions against misinformation by human fact-checkers. For example, Guess and Nyhan (2018); Vraga et al. (2020); Walter and Tukachinsky (2020) find evidence for the effectiveness of providing countermessaging; Lewandowsky et al. (2020) find that the countermessaging should explain *how* the misinformation came about, not just explain why the misinformation is false; and Druckman (2001); Hartman and Weber (2009); Berinsky (2017) find that the perceived credibility and affiliation of the countermessages matter a great deal. We recommend relying on such literature as a starting point to document the coherence of the chosen modeling means, application means, and ends.

### 7.5 Study effectiveness of chosen means

Our final recommendation is a call to action on research into the effectiveness of various automated fact-checking strategies. We have recommended relying on relevant literature to support connections between means and ends. However, for many proposed strategies, there is little or no literature. This represents a problem: if authors intend claims like "our system can help fight misinformation" to be scientific, they should provide (quantitative or qualitative) evidence for those claims. Where no prior literature exists on the efficacy of the proposed approach, authors can either rely on the expressed needs of people in their desired target groups (Nakov et al., 2021), or gather empirical evidence using a sample from the target group.

We highlight here the approach taken by Fan et al. (2020), where the authors demonstrated an increased accuracy of crowdworkers' veracity judgments when presented with additional evidence briefs from their system. This provides evidence for the effectiveness of their suggested narrative (assisted media consumption). Similar investigations were also done in Draws et al. (2022), where the authors explored crowdworker biases in fact-checking. Fruitful comparisons can also be made to evaluations of the effectiveness of explainability techniques (Vasconcelos et al., 2023), persuasive dialog agents (Tan et al., 2016; Wang et al., 2019; Altay et al., 2021; Farag et al., 2022; Brand and Stafford, 2022), and code assistants Perry et al. (2022). For fact-checking, this research should be informed by best practices on evaluating *human* interventions against misinformation (Guay et al., 2023), as well as best practises for human evaluations of complex NLP systems (van der Lee et al., 2019). Further, this research should carefully consider epistemic harms of the sort discussed in Section 6.5 before claiming superior performance.

## 8 Conclusion

Researchers working on automated fact-checking envision an epistemic tool for the fight against misinformation. We investigate narratives of intended use in fact-checking papers, finding that the majority describe how this tool will function in vague or unrealistic terms. This limits discussion of possible harms, prevents researchers from taking feedback from relevant stakeholders into account, and ultimately works against the possibility of using this technology to meet the stated goals.

Along with our analysis, we give five recommendations to help researchers avoid vagueness. We emphasise the need for clearly discussing data actors and application means, including groups beyond professional journalists as stakeholders, and ensuring that narratives are coherent and backed by relevant literature or empirical results. Although we have focused on automated fact-checking in this paper, we hope our recommendations can inspire similar introspection in related fields, such as hate speech detection.

## 9 Limitations

We extracted quotes only from the introductions and abstracts of the papers, following past findings that stories of intended use mostly occur in those sections (Birhane et al., 2022). Furthermore, we limited ourselves to clearly stated epistemic

elements, and did not account for any implied statement to avoid adding a layer of interpretation to the quotes. Therefore, we acknowledge the possibility of missing epistemic elements or narratives subtly stated or mentioned in other sections of the papers.

In addition, we analyse a list of 100 highly cited NLP papers focusing on automated fact-checking and related tasks. This list is non-exhaustive. Therefore, it certainly does not include all the influential papers, especially recent ones, as they may not have accumulated enough citations yet to surpass older work, which does represent a bias.

We chose to understand automated fact-checking artefacts through their intended uses. This is only one way understand technologies. As pointed out by Klenk (2021) it could be seen as somewhat simplistic, since artefacts are often appropriated for uses unforeseen by their designers. Artefacts may also hold very different properties depending on their sociotechnical contexts, i.e., the agents and institutions that may use or limit the use of the artefact (van de Poel, 2020). Recent work proposes to instead understand technological artefacts through their affordances (Tollon, 2022), i.e., the actions they enable (regardless of design). Our analysis focuses exclusively on the stated intentions of the authors, potentially limiting our findings. A complete analysis of the technology should also include how it is used in practice, e.g., of the documentation produced at companies, government agencies, and other groups that deploy fact-checking artefacts. We have left this for future work.

## 10 Ethical Considerations

As this is a review paper, we only report on publicly available data from the scientific publications shared in the supplementary materials. When reproducing quotes in our annotations, we did not anticipate any issues. However, if an author asks us to take any quotes from their paper down from our repository, we will do so.

## 11 Acknowledgements

We would like to thank Fabio Tollon, David Strohmaier, Hope McGovern, and Lucy Lu Wang for their helpful comments, discussions, and feedback. Michael Schlichtkrull is supported by the ERC grant AVeriTeC (GA 865958), Nedjma Ousidhoum was supported by the EU H2020 grant MONITIO (GA 965576) and Andreas Vlachos is supported by both AVeriTeC and MONITIO.

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

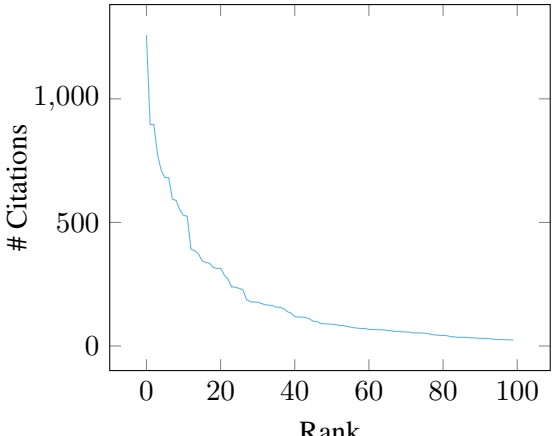

Figure 5: Number of citations per paper in our set, ordered according to rank.

# Appendix

## A   Providing veracity labels

Following our findings with respect to unsupported ends (see Section 6.3), we analysed the evolution over time of a particular case: providing veracity labels or scores to human fact-checkers as a means of limiting misinformation. Human fact-checkers have expressed strong skepticism at the usefulness of such scores for their work (Nakov et al., 2021). Thankfully, providing veracity labels to users in general is a declining trend. Organizing our data temporally (see Figure 7), we find that the percentage of papers proposing that users should act directly on model veracity prediction has decreased, whereas the percentage of papers proposing to identify potential claims for humans to fact-check, provide evidence for humans to act on, or actually replace human fact-checkers (i.e., write full articles discussing the evidence for and against claims) has increased.

## B   Annotation Guidelines

To study the epistemology of automated fact-checking ("AFC"), we annotate the narratives of 100 highly cited research papers.

We define a two-step annotation scheme: 1) a paragraph-level annotation and 2) a discourse-level narrative annotation. In the paragraph-level annotation, we extract quotes related to the goal and the methodology presented by identifying identify the a) data subjects, b) actors, c) model means, d) application means, and e) epistemic ends. Then, based on the identified elements, we extract the implied narratives in the discourse-level annotation.

| Year | 2008 | 2009 | 2014 | 2015 | 2016 | 2017 | 2018 | 2019 | 2020 | 2021 |
|---|---|---|---|---|---|---|---|---|---|---|
| # Papers | 1 | 1 | 1 | 4 | 5 | 11 | 23 | 15 | 28 | 11 |

Table 1: Papers per year in the sample of 100 highly cited papers we worked from.

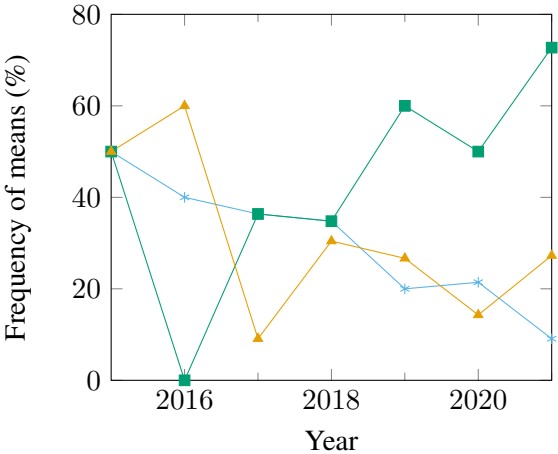

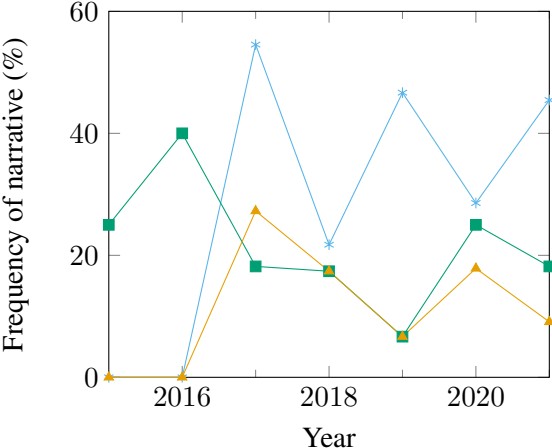

Figure 6: Evolution over time for providing veracity labels, other assistive means, and full automation. We see a steady decline in the percentage of papers expecting to directly present veracity labels to data actors. Note that the numbers do not add to 100%, as our annotation is multi-label.

We went through three-round annotation process. First, we annotated a small pilot set of five papers to define an annotation scheme. Then, working inductively, we annotated the full set of (100) research papers. That is, if, during the annotation, we encounter a means, end, actor, or narrative that does not fit any of our given categories, we introduce a new one. We then discussed our annotations and unified our set of introduced categories into those discussed in Appendixes B.2 and B.3. We further created a flowchart to help us move from paragraph-level to discourse-level annotation in a structured way (see Appendix B.5). Finally, we re-annotated all 100 papers based on our unified set of categories. The final set of criteria can be found in the sections below.

## B.1 Paper Selection

We collect research papers on automated fact-checking and related tasks (e.g., rumor detection). Working from the GitHub repository of fact-checking papers published alongside Guo et al. (2022)[3], we select the 100 most cited. We examine the introductions and abstracts of each paper to

---

[3]Check our GitHub repository for links to the papers.

Figure 7: Evolution over time for vague identification, vague opposition, and vague debunking.

extract quotes

## B.2 Paragraph-Level Annotation

We list the long-form definitions of epistemic elements we developed for consistent annotation here, along with the percentage of papers we found each element in. We include also category labels we expected based on our initial discussions, but which did not appear in our sample.

### B.2.1 Quotes

We extract the quotes/paragraphs from the introduction, yet, if a piece of information is missing, we further look at the abstract. We only examine epistemic quotes, i.e., related to knowledge. We sort them into quotes answering questions about narratives, i.e., the *why* and *what* of the paper.

### B.2.2 Data subjects

Based on *"who did what to whom for whom"*, we extract the subjects/people whose texts are fact-checked. They can be:

- journalists,
- citizen journalists,
- social media users,
- technical writers,
- public figures/politicians,

- product reviewers.

**Social media users (24.9%)** Covers any (potentially anonymous) contributors to social media as long as they are not explicitly public figures. This category includes people who comment on forum, Twitter/Facebook users, or editors of Wikipedia and other collaborative writing projects.

**Professional Journalists (15%)** Covers professional journalists, including fact-checkers, and anyone writing at a publishing house. This category also includes online publishers, but not collectives of amateur sleuths, e.g., Bellingcat. Organizations pretending to be journalists, e.g., satire sites and fake news websites, are also counted within this category.

**Citizen Journalists (0.4%)** Covers amateur journalists who take on the same work as professionals without funding or traditional education, e.g., bloggers, social media users, and collectives such as Bellingcat.

**Politicians & Public Figures (5.6%)** Covers any public figure, such as a politician or an actor. Analysed data could be media releases, interviews, speeches, or similar.

**Product Reviewers (1.7%)** Covers specifically product reviewers on sites such as Amazon, Trustpilot, where the purpose of the commenter is to describe and rate a product.

**Technical Writers (1.7%)** A few papers suggest scientists and other writers of technical documents should use fact-checking to, e.g., spot errors in their articles before publication. Along with scientific writers, this also covers writers of technical documents for areas such as business or health, as well as lawyers and clerks seeking to ensure consistency within legal documents.

### B.2.3 Data actors

The people who are supposed to **act** on the model outputs, such as journalists, social media moderators, or media users. From our data, they can be :

- professional journalists,

- citizen journalists,

- social media moderators,

- scientists,

- media consumers,

- technical writers,

- engineers and curators.

- law enforcement,

- algorithm.

**Professional Journalists (18.5%)** Covers professional journalists, including fact-checkers, and anything written at a publishing house. This category does not include amateur sleuths, e.g., Bellingcat, or bloggers doing the work of journalists.

**Citizen Journalists (2.6%)** Covers amateur journalists who take on the same work as professionals without funding or traditional education, e.g., bloggers, social media users, and collectives such as Bellingcat.

**Social Media Moderators (0%)** Covers people hired to moderate social media spaces. Applicable only when it is explicitly stated that a human employee should act on the model outputs.

**Scientists (3.9%)** Covers scientists, as well as any other actors who would use model outputs for scientific research. E.g., to analyse data with the express purpose of learning something about it, not acting on the model decisions.

**Media Consumers (4.7%)** Covers ordinary people who consume the content to which the fact-checking system is supposed to be applied. Only applicable when the consumer is directly understood (possibly implicitly) to use the system, e.g., in the form of a browser extension. The decision to use the outputs of the tool must be in the hands of the consumer.

**Engineers and Curators (1.3%)** Covers cases, where engineers or curators are maintaining a knowledge base of some kind, (e.g., Wikipedia), are intended to use the model outputs in their work.

**Law Enforcement (0.9%)** Covers cases where law enforcement agents, e.g., police officers, intelligence agents, or judges, are intended to act on the model outputs.

**Algorithm (1.7%)** Covers the cases of fully automated systems that act on the model outputs, e.g., remove posts based on the model's predictions.

### B.2.4 Model owners

The entities who **own** the models, or institutions who employ those who **act** on the model outputs. For instance, fully automated moderation systems are owned by the companies expected to use the models, e.g., social media companies. From our data, they can be :

- media companies,

- social media companies,

- law enforcement.

**Media companies (0.4%)**   Covers professional (non-amateur) media companies employing journalists and fact-checkers.

**Social Media Companies (4.7%)**   Covers social media companies more generally, including the engineers working to maintain the social network. Applicable when decisions will be made automatically based on the model's decisions, or when it is unclear.

**Law Enforcement (0.9%)**   Covers cases where law enforcement agents, e.g., police officers, intelligence agents, or judges, are intended to act on the model outputs.

**Scientists (1.3%)**   Covers scientists, as well as any other actors who would use model outputs for scientific research. E.g., to analyse data with the express purpose of learning something about it, not acting on the model decisions.

### B.2.5 Modeling (ML) Means

What concretely do the authors propose to do in terms of machine learning models? E.g., classifying claims, finding evidence, or similar. From our data, these can be:

- classify/score veracity,

- classify/score stance,

- evidence retrieval,

- produce justifications,

- corpora analysis,

- human in the loop,

- generate claims.

**Classify/score Veracity (79.3%)**   When the authors propose to classify the *veracity* of claims, i.e., whether or not the claim is true (or supported by evidence).

**Classify/score Stance (3.9%)**   When the authors propose to classify the *stance* of evidence, i.e., whether or not an evidence document takes a positive or negative view on a particular subject or claim.

**Evidence Retrieval (24.5%)**   When the authors propose evidence retrieval as a mean to reach their goal.

**Produce Justifications (6.9%)**   When the authors propose generating explanations/justifications to reach their goal.

**Human in the loop (11.2%)**   When there are human actors involved in the solution/main process described in the paper.

**Corpora Analysis (2.1%)**   When the authors propose using data analytics or any sort of corpora analysis in their methodology.

**Generate claims (0.9%)**   When the authors propose generating claims, e.g., to produce additional misinformation to train on.

### B.2.6 Application Means

How do the authors want to use what is developed in the paper to accomplish a specific goal (e.g., reduce the spread of misinformation)? For example, this could be by deploying automated systems to show social media users content warnings about claims which might be false (along with, potentially, evidence for their falsity). From our data, we identify the following suggestions:

- identify claims,

- triage claims,

- supplant human fact-checkers,

- gather and present evidence,

- identify multimodal inconsistencies,

- automated removal,

- provide veracity labels/scores,

- provide aggregates of social media comments,

- filter system outputs,

- maintain consistency with KB,

- analyse data,

- produce misinformation,

- vague persuasion.

**Identify claims (26.2%)** There are many claims on the internet and most of them are not misinformative. ML should be deployed to find the misinformative ones.

**Triage claims (6.0%)** Fact-checkers, content moderators, and similar have many claims to deal with. ML should be deployed to rank them, so more costly actors focus on the most important ones first.

**Supplant Human fact-checkers (13.7%)** Replace human fact-checkers entirely, or at least partially by automatically handling some claims (in fully automated fashion). If the intention is a human-in-the-loop system, *supplanting human fact-checkers* is not the means.

**Gather and present evidence (10.7%)** An ML model should find relevant evidence supporting/refuting a claim, and show it to a human. It can also involve generating justifications for how the evidence relates to the claim.

**Identify multimodal inconsistencies (1.7%)** For multimodal misinformation, an important tool identifies mismatches between modalities. ML models should do this, then show the results to humans.

**Automated removal (1.7%)** ML models should automatically remove claims from e.g., social media platforms, with no human involvement.

**Provide veracity labels/scores (17.5%)** ML models should provide indications of truth value to data actors, either in the form of labels or veracity scores.

**Present aggregates of social media comments (6.9%)** ML models should aggregate the stance/evidence presented in the comments to a potentially misinformative claim, as a way of summarizing points made by humans in favor of or against the claim.

**Filter system outputs (3.0%)** Other NLP/ML models, e.g. LLMs, struggle to produce truthful outputs. Fact-checking models should filter or re-rank their output. This also includes extractive models, e.g., relation extraction systems, that are coupled with a fact-checker to only return (or add to a KB) things the fact-checker accepts as truthful.

**Maintain consistency with KB (1.3%)** Internal knowledge bases (defined loosely, including textual ones, such as Wikipedia) can be used as a source of truth to prevent the data actor from publishing untruths. This could be at writing time, e.g., as a writing assistant, or it could be by continuously keeping an article published online up to date.

**Analyse data (3.0%)** Fact-checking models and datasets should be used for research purposes, e.g., to analyse misinformative text to get a better understanding of how misinformation spreads.

**Produce misinformation (0.9%)** A machine learning model that produces misinformation can be used *against* misinformation, for example, to generate adversarial data, or to show people what such models are capable of to innoculate them against it when they encounter it in the wild.

**Vague persuasion (3.4%)** ML models should somehow convince people to change their opinions on claims, e.g., by providing warning labels or presenting evidence. The mechanism is *not specified*, though.

### B.2.7 Ends

The purpose or ultimate aim of the approach. What do the authors want to accomplish? From our data:

- limit misinformation,

- limit AI-generated misinformation,

- increase the veracity of published content,

- develop knowledge of NLP/language,

- avoid biases of human fact-checkers.

- detect falsehood for law enforcement

**Limit misinformation (78.1%)** The ultimate aim of the paper is to prevent misinformation from spreading or to limit its influence.

**Limit AI-generated misinformation (3.4%)** The ultimate aim of the paper is to prevent AI-generated misinformation from spreading or to limit its influence.

**Increase veracity of published content (7.2%)**
Some systems are intended to be used by publishers and writers before their content is made public – or applied to continuously keep published content up to date. The aim here is not to limit already spreading misinformation but to keep people from accidentally publishing untrue things. Published content can be small data, like a single article, a large collection of data, like Wikipedia, or a knowledge base.

**Develop knowledge of NLP/language (3.4%)**
Automated fact-checking is a difficult problem. By studying it, we can learn more about how language works, and how to build models that interpret semantics.

**Avoid biases of human fact-checkers (0.4%)**
I.e., develop "super-human" fact-checking.

**Detect falsehood for law enforcement (0.9%)**
One proposed use case for fact-checking and similar technologies is as truth-telling systems for law enforcement, e.g. in courtrooms.

### B.2.8   Important Note

In this annotation stage we do not extract implied statements, we rely directly on extracted quotes and do not interpret them.

### B.3   Narratives

At the discourse level, we extract the epistemic narratives present in the paper. We envision narratives as discourse structures combining the (modeling/application) means, ends, data actors, and data subjects extracted at the paragraph level. Note that we use fictional examples below to avoid biasing annotators' decisions on specific papers. The narratives can be:

- vague identification,

- vague debunking,

- vague opposition,

- assisted content moderation,

- automatic content moderation,

- assisted media consumption,

- assisted internal fact-checking,

- assisted external fact-checking,

- automated external fact-checking,

- assisted knowledge curation,

- scientific curiosity.

- truth-telling for law enforcement

- vague moderation,

- adversarial research.

We do not account for implied paragraph-level narratives but only for annotated elements that *are* in the quotes. As we extract multiple quotes, papers may have more than one narrative present.

**Vague Identification (31%)**   The paper mentions identifying or detecting misinformative claims as the means, and limiting misinformation as the end. However, it is not clear how the authors intend to accomplish that end using those means. Typically, there are no data actors – it is also not clear who should act on the model's predictions.

*Vague identification* Applies only in cases where the authors say that they want to identify or detect misinformation without saying what they are going to do with that afterward, e.g., in rumor detection papers where they claim that they want to detect rumors but we do not know what they want to do with this identification or classification labels. A fictional example follows below:

*"Misinformation is a major societal problem, eroding community trust and costing lives by e.g., inducing hesitance to adopt life-saving vaccines. The amount of messages spread on social media has increased drastically in recent years. Therefore, it is necessary to automatically identify potentially misinformative claims in order to address this problem."*

**Vague Debunking (14%)**   The authors propose to produce evidence-based debunkings of text via automated means, but it is not clear whether the introduced model is an assistive tool for fact-checkers or fully automated. Further, it is not clear how the "debunkings" will be communicated to misinformation-believers.

It is clear that the mechanism is supposed to follow what fact-checkers are currently doing, but not where or how the ML model will be used in this process. Furthermore, it is *unclear* whether the entire process will be automated. For example, the paper may suggest that an automated fact-checking model should be used by fact-checkers, but it is not clear *how* the model assists in that. A fictional example follows below:

| Category Label | Description | % |
|---|---|---|
| Vague identification | Identify misinformation without explaining what happens afterwards. | 31 |
| Automated External Fact-checking | Fully automate the work of journalistic fact-checkers. | 22 |
| Vague opposition | Opposition to misinformation with no clear application means. | 19 |
| Assisted External Fact-checking | Assist journalists writing fact-checking articles e.g., by finding evidence. | 18 |
| Vague Debunking | Evidence-based debunking with no clear users. | 14 |
| Assisted media consumption | Assist media consumers by e.g., providing extra information. | 8 |
| Scientific curiosity | Develop AFC artefacts to learn something about e.g., language or NLP. | 8 |
| Assisted Knowledge Curation | Assist humans in curating public knowledge bases, e.g., Wikipedia. | 7 |
| Assisted internal fact-checking | Use AFC in journalistic contexts to fact-check before publication. | 4 |
| Automated content moderation | Fully automate content moderation on social media platforms. | 4 |
| Law enforcement | Use AFC as a truth-telling system in law enforcement contexts. | 1 |

Table 2: Epistemic narratives found in our analysis of 100 fact-checking papers. Along with each narrative, we list the percentage of papers in which we found it.

E.g., *"Misinformation is a major societal problem, eroding community trust and costing lives by e.g., inducing hesitance to adopt life-saving vaccines. One proposed solution is to debunk circulating claims, i.e. to find evidence against them. This is the process commonly carried out by fact-checking organizations, e.g. PolitiFact. In this paper, we introduce a classifier..."*.

**Vague Opposition (19%)** Restricted to cases without any application means (model means and no application means in contrast to vague identification and vague debunking). E.g., *a machine learning/automated system will reduce the spread of misinformation.*

When the paper presents a narrative of vague opposition to misinformation. The *end* is to limit the spread or influence of misinformation, and the *ML means* are, for example, to classify claims. However, the connection between means and ends is left unmentioned, and epistemic actors are typically absent. An impression is given that the development of automated fact-checking will limit the spread of misinformation, but the link between the two is left unstated. A (fictional) example follows below:

*"Misinformation is a major societal problem, eroding community trust and costing lives by e.g., inducing hesitance to adopt life-saving vaccines. It is therefore of paramount importance that the spread of false information is stopped. Automated fact-checking – that is, the automatic classification of claim veracity – represents one solution to this critical problem."*

**Assisted Content Moderation (0%)** If the paper proposes the deployment of automated fact-checking as a tool to assist content moderators on social media platforms. Here, the *end* is to limit the spread of misinformation on social media plat-

forms, the *means* is to provide suggestions for posts to delete (along with, potentially, evidence for why they might be false), and the *actors* are human moderators who make the final choice on whether posts should be deleted or not. A (fictional) example follows below:

*"Social media is rife with misinformation, eroding community trust and costing lives by e.g., inducing hesitance to adopt life-saving vaccines. One solution is for moderators to remove information deemed false. However, with the number of posts made every day on social networks, this strategy is too costly. In this paper, we develop an automated system for filtering claims, helping moderators quickly discover and make decisions on circulating claims."*

**Automated Content Moderation (4%)** If the paper analysed proposes a similar content moderation strategy, but instead of assisting human moderators, it suggests replacing them entirely. In this case, the *end* is to limit the spread of misinformation on social media platforms, the *means* is to deploy classifiers to truth-tell claims and remove any labeled false, and the *actors* are the algorithm (as well as, implicitly, the model owners – the executives and engineers at social media companies who deploy and make decisions about such systems). A (fictional) example follows below:

*"Social media is rife with misinformation, eroding community trust and costing lives by e.g. inducing hesitance to adopt life-saving vaccines. One solution is for moderators to remove information deemed false. However, with the number of posts made every day on social networks, this strategy is too costly. In this paper, we develop an automated system for detecting false claims, which can serve as a first line of defense against misinformation."*

**Assisted Media Consumption (8%)**  If the paper proposes to deploy automated fact-checking as an assistive tool for *consumers* of information, either as a layer adding extra information to social media posts or as a standalone site where claims can be tested. In this case, the *end* is either to limit the influence of misinformation or to induce veracious mental states in users; the *means* is to deploy automated systems to warn about claims which might be false (along with, potentially, evidence for why their falsity); and the *actors* are social media users or information seekers in general. The actors could also be social media companies who show information produced by fact-checking assistants to users as an integral part of their UI. A (fictional) example follows below:

*"Misinformation is a major societal problem, eroding community trust and costing lives by e.g. inducing hesitance to adopt life-saving vaccines. It is therefore of paramount importance that the spread of false information is stopped. Studies have shown that many people adopt beliefs without doing due diligence on the information they receive. Automated fact-checking – that is, the automatic classification of claim veracity – could via e.g., a plugin be deployed to warn social media users about potentially false claims."*

**Assisted Internal Fact-checking (4%)**  When the paper proposes to deploy automated fact-checking as an assistive tool for journalists, deployed internally. Here, the *end* is to increase the veracity of published information, the *means* is to deploy automated systems to warn about claims which might be false (along with, potentially, evidence for why their falsity), and the *actors* are either journalists employed at traditional publishing houses or citizen journalists. A (fictional) example follows below:

*"Research is a fundamental task in journalism, conducted to ensure published information is truthful and to protect the publisher from libel suits. This is a crucial step, which journalists – strained by the advent of the 24-hour news cycle – increasingly skip. Given a trusted source of evidence documents, such as LexisNexis, much of the grunt work of research could be handled by automated fact-checkers, leaving journalists free to tackle the hardest parts, e.g. double-checking information with sources."*

**Assisted External Fact-checking (18%)**  When the paper proposes to deploy automated fact-checking as an assistive tool for journalists, deployed for external fact-checking. Here, the *end* is to limit the influence of misinformation, the *means* is either to speed up the production of counter-messaging by surfacing relevant evidence or to direct journalists to the most problematic currently circulating claims, and the *actors* are either journalists employed at fact-checking organizations such as Full Fact or citizen journalists.

This is restricted to when the paper proposes to improve one or multiple components in the automated fact-checking pipeline rather than the whole pipeline/end-to-end system (in the latter case it becomes *automated external fact-checking*, see following paragraph). I.e., when it is human-in-the-loop, then it is assisted fact-checking but not automated.

A (fictional) example follows below:

*"Misinformation is a major societal problem, eroding community trust and costing lives by e.g., inducing hesitance to adopt life-saving vaccines. An important way to fight misinformation is the production of relevant counter-messaging, i.e., the work done by organizations such as Full Fact or PolitiFact. With the number of false claims published on social media every hour, it is not feasible for human journalists to debunk them all. Journalists could use automated fact-checking to triage incoming claims to limit the workload, or to quickly surface relevant evidence while producing articles."*

**Automated External Fact-checking (22%)**  When the paper proposes to fully automated external fact-checking, including all parts of the pipeline. *end* is to limit the influence of misinformation, the *means* are to entirely replace human fact-checkers by automatically producing fact-checking articles, and the *actors* are users engaging with the produced content. It is only automated external fact-checking when the authors *explicitly* say the process should be automated, otherwise it is vague debunking (see previous paragraph).

A fictional example can be seen below:

*"Misinformation is a major societal problem, eroding community trust and costing lives by e.g., inducing hesitance to adopt life-saving vaccines. An important way to fight misinformation is the production of relevant counter-messaging, i.e., the work done by organizations such as Full Fact or*

*PolitiFact. With the number of false claims published on social media every hour, it is not feasible for human journalists to debunk them all. As such, the process must be automated."*

**Assisted Knowledge Curation (7%)** When the paper proposes fact-checking primarily as a component filtering the information kept in some curated knowledge vault, including graph-based knowledge bases as well as text-based collections such as Wikipedia. Here, the *end* is to increase the veracity of the knowledge vault, the *means* is to use automated fact-checking as an additional truth-telling layer that prevents disputed facts from being added (or interrogates already added facts), and the *actors* are typically the engineers who maintain the knowledge base. A (fictional) example follows below:

*"Knowledge bases fuel many real-world NLP applications, e.g., question answering. The maintenance of knowledge bases is an expensive process, yet as new facts appear in the world knowledge bases must be kept up-to-date. Automated triple extraction from e.g., news data has been proposed as an alternative to human annotators; yet, the quality remains low. Automated fact-checking systems, which verify facts against trusted knowledge sources, could be used to prevent highly disputed facts from being entered – or ensure that new facts are consistent with the existing knowledge."*

**Truth-telling for Law Enforcement (1%)** When the paper proposes fact-checking primarily as a lie detector for use in law enforcement contexts. This could include police work as well as courtroom applications. A fictional example can be seen below:

*"Automatic detection of deception is commonly used in police work via e.g. polygraphs. However, accuracy remains low. We propose that automated fact-checking via NLP could represent a viable alternative."*

**Scientific Curiosity (7%)** When the authors of the paper justify their projects purely based on scientific curiosity. While differing strongly from the other narratives presented here, this is still a virtue epistemic narrative, concerned with the production of *good knowledge*. Here, the *end* is to increase scientific knowledge of semantics; the *means* is to learn how to build automated systems that mimic human fact-checkers, a process theorised to yield knowledge about the construction of meaning; and the *actors* are scientists in natural language pro-

cessing and adjacent fields. A (fictional) example follows below:

*"Journalistic fact-checking is a difficult task, requiring reasoning about disputed claims that fool sufficiently many humans to warrant professional attention. For systems to mimic fact-checking to a substantial degree, significant semantic understanding is necessary. As such, automated fact-checking is an ideally suited field to develop and test new models for natural language understanding."*

### B.4 Feasibility Support

We annotate narratives with any support for their feasibility given in the paper. Vague narratives are by definition not supported – if the narrative is clear enough to design e.g., a user study to test if the means are an effective way to reach the end, it is not vague. We use the following categories:

**Scientific Research** The feasibility of the narrative is supported by reference to scientific studies. This only applies if the entire narrative is supported, i.e., a scientific paper is supported for the means being a feasible way to reach the end. This Does NOT apply if only e.g., the dangers of misinformation are supported – the proposed means being an effective strategy MUST be entirely supported.

For example, the crowdworkers' study in the FactCheckingBriefs (https://aclanthology.org/2020.emnlp-main.580/) could be cited to demonstrate that giving people evidence increases their accuracy on veracity judgments. This applies even if the cited paper does not support what the paper claims it supports, e.g., if someone cites the FEVER dataset (Thorne et al., 2018) for fully automated fact-checking being an effective strategy to reduce the spread of real-world misinformation. This almost always applies to narratives of scientific curiosity, as researchers tend to show their research is a useful strategy for studying what they investigate by writing a related works section.

**News Media** The feasibility of the narrative is supported by a reference to a newspaper article. This occurs in the same cases as scientific research, except the citation is to a newspaper article rather than a scientific article.

**Automation** The feasibility of the narrative is supported by reference to the studied task currently working as a human task. It is assumed that full

or partial automation will work, and will be effective – i.e., no reference to studies testing whether the proposed strategy helps humans in the loop is given.

**Example** *"While the number of organizations performing fact-checking is growing, these efforts cannot keep up with the pace at which false claims are being produced, including also clickbait (Karadzhov et al., 2017a), hoaxes (Rashkin et al., 2017), and satire (Hardalov et al., 2016). Hence, there is need for automatic fact checking."* (Baly et al., 2018)

**Vague Community** The feasibility of the narrative is supported by reference to the preference of practitioners, usually without citing a scientific paper. This includes phrases like *"many people think this is a good idea"*.

**Example** *"Many favor a two-step approach where fake news items are detected and then countermeasures are implemented to foreclose rumors and to discourage repetition."* (Potthast et al., 2018)

**Sense of Threat** The feasibility of the narrative is not directly supported. However, the paper argues that the strategy represented by the narrative (e.g., automated content moderation) must be done, or bad things will happen. The bad thing is something other than human workers not being able to keep up with their workload, as the narrative otherwise falls under automation.

**Example** *"An abundance of incorrect information can plant wrong beliefs in individual citizens and lead to a misinformed public, undermining the democratic process. In this context, technology to automate fact-checking and source verification (Vlachos and Riedel, 2014) is of great interest to both media consumers and publishers."* (Yoneda et al., 2018)

### B.5 Paper Metadata

**The Type of the Paper** Survey, dataset, model description, task description, other (name).

**The Subfield/task** Can be fact-checking, rumor detection, misinformation, disinformation, or other (name the task).

**The Subtask** The part of the pipeline in which the model interacts, e.g., claim detection, evidence retrieval, verdict prediction, justification production.

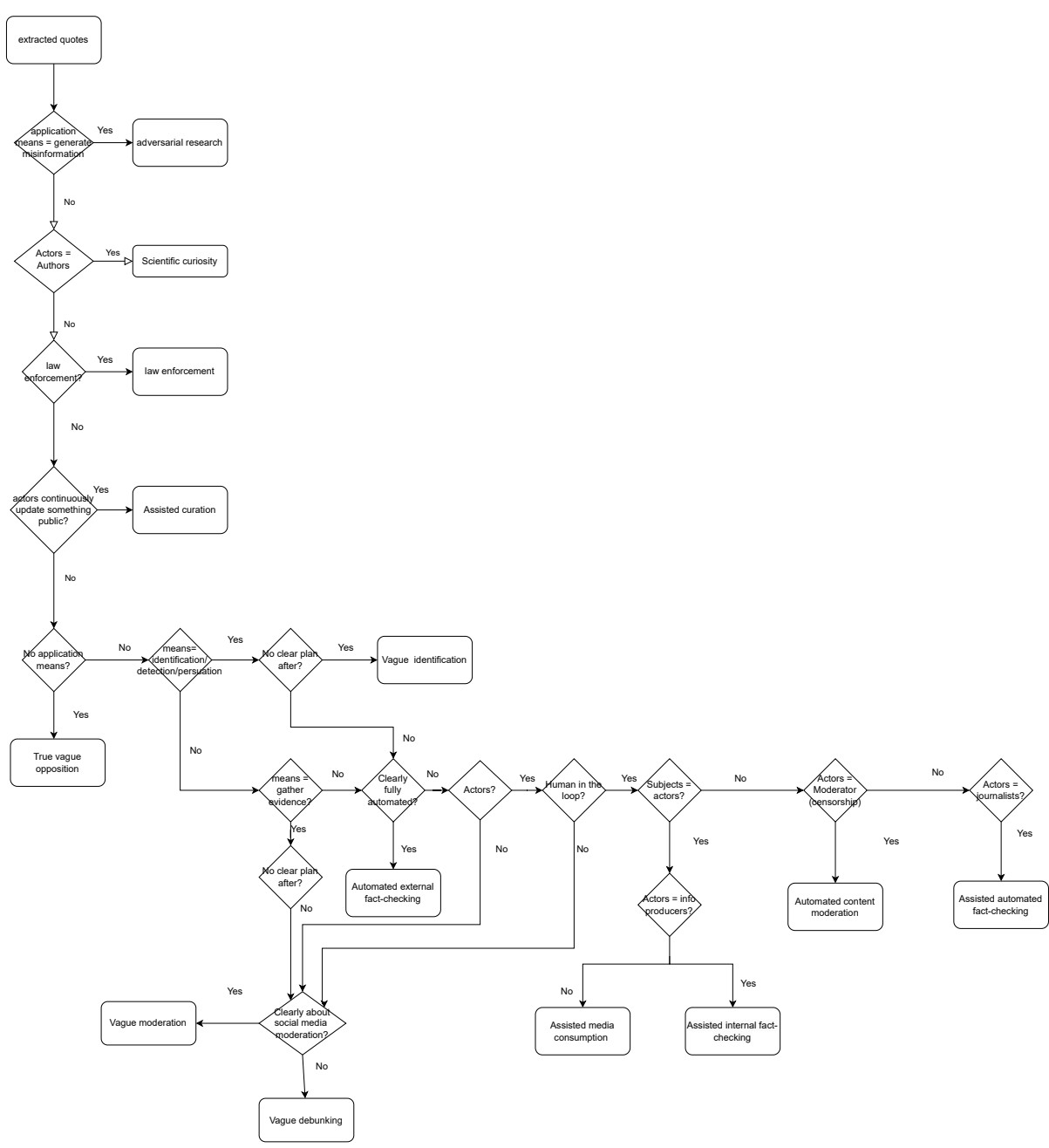

Figure 8: Flowchart of the Narrative Annotations.

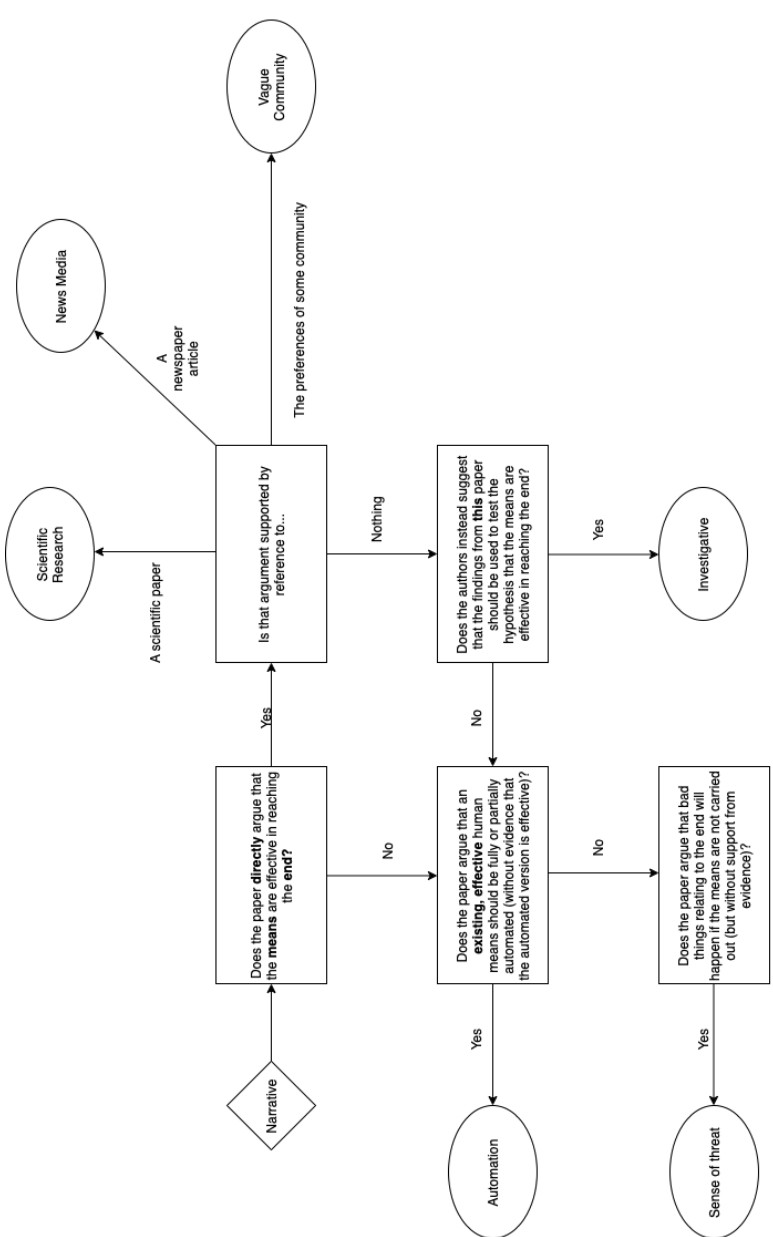

Figure 9: Flowchart of the Feasability Support Annotations.