# OpenReview forum: "The Intended Uses of Automated Fact-Checking Artefacts: Why, How and Who"
_EMNLP/2023/Conference — EMNLP 2023 Findings_

### Official Review · Reviewer_XbZ4 · 2023-07-22

**Soundness:** 4

**Excitement:**

5: Transformative: This paper is likely to change its subfield or computational linguistics broadly. It should be considered for a best paper award. This paper changes the current understanding of some phenomenon, shows a widely held practice to be erroneous in someway, enables a promising direction of research for a (broad or narrow) topic, or creates an exciting new technique.

**Paper Topic And Main Contributions:**

This review/position paper identifies common pitfalls in fact checking papers, e.g., vagueness about who should use the proposed systems and how. A comprehensive categorization of such issues is proposed and applied to the analysis of the 100 most highly cited papers in the area, using a content analysis approach conducted by the authors. Finally, recommendations are provided for improving narratives in fact checking contributions.

While I have not worked on fact checking myself, I am familiar with the literature and the overall rationale.

**Questions For The Authors:**

A: For unsupported ends, the existence of counter evidence is given as a sufficient criterion. Is it, in general, a criterion for unsupported ends? What if there is both evidence and counter evidence? This black and white view of fact checking may be an oversimplification. Have you considered a more nuanced categorization of faithfulness to evidence, e.g. claim strength as in https://aclanthology.org/2021.emnlp-main.845/ ?

**Reasons To Accept:**

Important examination of the very basic goals of automated fact checking research and the extent to which they are expressed in influencial publications in the area, finding low coherence in many cases.

Clear, well motivated and systematic argumentation that uses just the terminology and reasoning strategies common in the fact checking literature, e.g., lacking evidence, making it well positioned toward this community.

Robust and thorough annotation and analysis procedure following best practices in content analysis, resulting in both a novel categorization scheme and annotated corpus used to quantitatively back up the arguments in the paper. Exceptionally detailed account of the annotation guidelines and the meaning of each label, including examples for each single case.

Broad and informative perspective on related work on the goals of fact checking from various disciplines.

**Reasons To Reject:**

Seems to be implicitly making a strong assumption, that (influencial) papers in automated fact checking always contribute an artifact positioned within some (potential) application. Contributions in NLP, on the other hand, often focus on improving or investigating specific aspects of components that can be internal to such a system, leaving out a broader discussion on the overall system, especially given space limitations. It is therefore not entirely clear whether the normative statement the authors are making, that a comprehensive narrative is always warranted, is a subjective preference or if there is evidence that it is necessary in all cases. For example, arguably it is not the researcher's job to discuss what to do with the identified misinformation after it was found by the system.

No information on the papers themselves is given in the paper, such as distribution over publication years or number of citations. It is also not specified how the number was obtained.

**Reproducibility:**

4: Could mostly reproduce the results, but there may be some variation because of sample variance or minor variations in their interpretation of the protocol or method.

**Reviewer Confidence:**

3: Pretty sure, but there's a chance I missed something. Although I have a good feel for this area in general, I did not carefully check the paper's details, e.g., the math, experimental design, or novelty.

**Typos Grammar Style And Presentation Improvements:**

Feasibility support does not seem to appear in figure 1. This is fine, but may be worth mentioning so the reader does not look for it.

Citing Spinoza and Heidegger seems somewhat out of context given the subject of technological artifacts, which is implicitly about information technology. Expanding the discourse more explicitly may help.

---

> ### Author Rebuttal · Authors · 2023-08-28
>
> We thank the reviewer for their positive comments, and we are very happy they found our paper important and well-positioned towards EMNLP!
>
> **Assumptions about Author Intent**
>
> We thank the reviewer for pointing this out. This is a great point! We do not make a strong statement that automated fact-checking papers \*always produce\* such an artefact. However, authors of fact-checking papers often do. As we mention in lines 487-491:  *“Authors of fact-checking papers clearly believe their research to be solving a societal need: 82% of narratives in our analysis had ‘limiting misinformation' as the desired end, and an additional 7% had ‘increasing the veracity of published content’”*. The reviewer is correct that authors may have other motivations. For example, the “scientific curiosity” narrative (which appears in 8% of the papers in our analysis) is often not directly related to fighting misinformation.
>
> We argue that if authors wish to make such claims in their papers, they should make sure there is empirical support, either by gathering evidence themselves or by citing other work that presents evidence. Unfortunately, as our paper shows, this rarely happens.
>
> We appreciate that our findings on the prevalence of such arguments come somewhat late in the paper, appearing first in the section header for Section 7. We will bring these to the introduction in the camera-ready version if our paper is accepted.
>
> **List of papers**
>
> We included the URL links of the papers that we analysed in the supplementary materials (the data.json file with the extracted quotes). If our paper is accepted, we will add the list to the camera-ready. The full JSON file with our annotations will also be made available on GitHub.
>
> **Analysis Statistics**
>
> We thank the reviewer for pointing this out, we agree statistics on citations and publication year would be useful. Number of citations ranges (according to Google Scholar) from 1258 to 16. We will include a figure showing this in the camera-ready if our paper is accepted. As for the number of papers per year, we will also add the following table:
>
> | Year     | 2008 | 2009 | 2014 | 2015 | 2016 | 2017 | 2018 | 2019 | 2020 | 2021 |
> |----------|------|------|------|------|------|------|------|------|------|------|
> | # Papers | 1    | 1    | 1    | 4    | 5    | 11   | 23   | 15   | 28   | 11   |
>
> **Figure 1**
>
> Thanks for pointing this out – we will amend the figure.
>
> **Spinoza and Heidegger**
>
> Spinoza and Heidegger both made arguments that artefacts should be understood as functions of their uses. Spinoza argued that a hatchet which does not work for its intended purpose – chopping wood – is no longer a hatchet. Heidegger went further, arguing that artefacts can only be properly understood when actively used for their intended purpose; i.e., the only way to understand a hammer is to hammer with it. We use the citations to document that some philosophers hold uses to be important for the understanding and analysis of artefacts. We will expand on this in the camera-ready if our paper is accepted.

---

### Official Review · Reviewer_gXnU · 2023-07-28

**Soundness:** 4

**Excitement:**

4: Strong: This paper deepens the understanding of some phenomenon or lowers the barriers to an existing research direction.

**Paper Topic And Main Contributions:**

The authors investigate the usefulness of research papers about automated fact-checking. They argue that their usefulness decreases without considering who will be using the developed fact-checking artifacts, and without specifying how they should be used. Therefore, they identify relevant components of a fact-checking system and document their degree of consideration in 100 popular fact-checking papers.
They contribute a holistic perspective on fact-checking systems, critically analyze a high number of existing papers, define influential components for the usefulness of fact-checking systems, and derive recommendations for future research. They document the results of the analyzed papers in the form of annotations, which is provided in a json file for further analysis.

**Questions For The Authors:**

The numbers in "()" refer to the page numbers in the paper.

(1)
Where does the term "artifacts" come from?

(3)
Put the epistemic elements and narrative types in a table along with short descriptions and simple examples. This can also be part of the appendix if too large for the body.

(3)
There are many percentages given for epistemic elements and the narrative types, and many of them do not make sense to me. E.g., the given categories for "Model Owners" total in 8% - what about the other 92%? And the categories for "Modeling means" exceed 100% - how can that be? And the "Vague narrative" types should make up 56% (as stated in the paper (on page 4), but the percentages add up to 64%? There are more examples like this in the paper - make sure to correct them, or to clarify how the percentages relate.

(4)
What's the discourse level?

(4)
What is the difference between vague debunking and vague opposition?

(6)
Chapter "Evidence is not a silver bullet" is well researched and the arguments made in it are all valid. However, the paper is about improving the usefulness of fact-checking related research, and stating these general psychological phenomena which cannot be prevented by fact-checking systems is misplaced here, in my opinion. You should at least give suggestions on how researchers can consider these points in their work.

(7)
One point in your recommendations is to include the data actor in fact-checking papers. It totally makes sense to think from the perspective of the target group when developing a tool, but do you have any suggestions of how to involve them? Are you suggesting to actually invite a sample from the target group as evaluators? How are you going to know what they need if they have a special role which you are not familiar with?

(15)
"Social Media Moderators (0%)" ??

**Reasons To Accept:**

The paper is well written and structured. The research was conducted systematically and logically, and the arguments made are thorough and valid. When accepted, the paper will draw researchers' attention towards the usability of fact-checking systems. It might especially influence those who "only" develop a technological solution without considering its practical and societal benefit. The paper would also introduce a new paradigm for evaluating the usefulness of fact-checking papers and systems. It would raise awareness of important aspects to consider for all those who work on fact-checking systems and care about its value for the community.

**Reasons To Reject:**

There are many categories and new terms introduced which are hard to distinguish without concrete examples. The wording of those terms is also somewhat confusing, so that it is hard to read the paper at the first time. Especially when there are terms with subtle differences, like many of the sub-categories for epistemic elements and narrative types in the paper, combining a few of the terms would have been a good idea in order to make those rememberable for the reader.
One might argue that many of the analysis results and recommendations are quite simple and intuitive. It might have been more useful to find out why the analyzed papers have those shortcomings of usefulness instead of supporting the fact that they do have them. In my opinion the authors should have addressed the "why" and at least come up with ideas for follow-up research.

**Reproducibility:**

5: Could easily reproduce the results.

**Reviewer Confidence:**

4: Quite sure. I tried to check the important points carefully. It's unlikely, though conceivable, that I missed something that should affect my ratings.

**Typos Grammar Style And Presentation Improvements:**

(1)
"Connecting research to potential use allows researchers to shape their work taking into account the expressed needs of key stakeholders, such as professional fact-checkers (Nakov et al., 2021)." --> Overcomplicated sentence:

(2)
"In this paper, we investigate narratives about intended use in automated [...]" --> "uses" or "the intended use"

(2)
"Based on this, we give recommendations to clarify discussions of intended use:" --> weird style

(2)
"[...] time, rendering resulting artefacts diffi- 122
cult to use in real-world scenarios." --> style

(8)
"We investigate narratives of intended use in fact-checking papers, finding that the majority describe how this tool will function in vague or unrealistic terms" --> "[...] papers, and find that [...] how a tool [...]."

(9)
"We chose to understand automated fact-checking artefacts through their intended uses. This is only one way understand technologies." --> "[...] one way to understand technologies."

(23)
Figure 7 should be formatted in a cleaner way.

---

> ### Author Rebuttal · Authors · 2023-08-28
>
> We thank the reviewer for their positive comments. We are thrilled they find the paper well-written, thorough, and valid. We hope the reviewer is right that the paper will influence authors to give stronger consideration to the practical and societal impacts of their technological artefacts!
>
> **Terminology – Definitions & Examples**
>
> We thank the reviewer for highlighting their clarity concerns. We would appreciate it if the reviewer could specify which terms they find confusingly worded so that we can improve our definitions.
>
> We do include simple definitions for each narrative and epistemic element, albeit in Appendices B.2 and B.3 – as the reviewer points out, this is a long list and as such was moved from the main body of the paper. We also include an example for each narrative in Appendix B.3, as well as a table like the one requested by the reviewer (Table 1). If accepted, in the camera-ready we will move Table 1 to the main text, and add an example for each epistemic element to the appendix.
>
> **Intuitive Findings**
>
> We agree that our findings and recommendations seem quite intuitive. However, we do not know of any previous paper that studies this question or provides recommendations similar to ours. Documenting that the phenomenon exists and making the community aware of the problem is (hopefully) the first step to fixing it.
>
> **Follow-up Research**
>
> We agree that answering why so much vagueness occurs is an interesting question. We are happy to theorise, and will include a section on this in the camera ready if our paper is accepted; perhaps authors simply prefer leaderboarding to evaluation that includes data actors? Perhaps they believe questions about effectiveness to be obvious – “of course automatically identifying potential misinformation will help us fight it”? However, without e.g., interviewing authors, we will have to hedge this as we do not have access to their internal thought processes as they are writing their papers.
>
> We do believe this is an interesting direction for future research. Producing a scientific answer (rather than just a guess) would include for example ethnographic research with authors. [2] carried out such analysis with high-energy physicists and molecular biologists – we would find a similar study with automated fact-checking authors (or NLPers in general) utterly fascinating.
>
> [2] Epistemic Cultures: How the Sciences Make Knowledge. Knorr-Cetina, 1999.
>
> **Questions**
>
> _(1) Where does the term "artifacts" come from?_
>
> “Artefacts” is a common term in philosophy referring to deliberate constructs, originating from Aristotle – see https://plato.stanford.edu/entries/artifact. With that said, we actually picked up the term from the ARR Responsible NLP Checklist (https://aclrollingreview.org/responsibleNLPresearch/).
>
> _(3) Put the epistemic elements and narrative types in a table..._
>
> See section on terminology above.
>
> _(3) There are many percentages given for epistemic elements..._
>
> Our annotation scheme is multi-label, as stated in lines 158-159: “This annotation is multi-label, i.e., for each element, one quote may have several (or no) labels.” This means percentages do not sum to 1. For example, a paper may include both “classify/score veracity” and “evidence retrieval” as modelling means, if e.g. the system introduced in the paper is a pipeline. We will make this clearer in the camera-ready if our paper is accepted.
>
> _(4) What's the discourse level?_
>
> Defined on line 177: “Annotation at the discourse level is annotation spanning multiple quotes.” For example, the narrative shown in Figure 1 includes an epistemic end from one quote, and a modelling means from another. This means the annotation is at the level of “discourse” (as opposed to paragraph/quote).
>
> _(4) What is the difference between vague debunking and vague opposition?_
>
> Vague opposition: No discussion in the paper of how the developed artefacts will be used to fight misinformation (although it is claimed that fighting misinformation is the goal). Typically: no application means.
>
> Vague debunking: The paper claims the developed artefacts will assist in the production of evidence-based refutations, and that this will fight misinformation. However, there is a missing link between producing debunkings and using them. Typically: no data actors.
>
> See also lines 271-286, as well as the flowchart in Figure 7 on p23.
>
> _(6) Chapter "Evidence is not a silver bullet" is well researched..._
>
> The “evidence is not a silver bullet”-section was intended to give a recommendation to authors, albeit implicitly. As we mention, the common solution given in papers to counteract the effect of bias in models is to rely on evidence. We argue that evidence does not present a complete solution (i.e., not a silver bullet) for this problem. As such, authors should be careful to make such claims. We will make this clearer in the camera-ready if our paper is accepted.
>
> _(7) One point in your recommendations is to include the data actor..._
>
> We are indeed suggesting that evaluating based on the preferences and needs of the target user group (i.e. fact-checkers, media consumers, …) might be preferable to leaderboarding. This could include explicit involvement of people from that group, yes. We understand that this can be difficult in practice. We will develop this further in the camera-ready if accepted. Ideally, if authors intend claims like "our system can help fight misinformation" to be scientific, they should have empirical evidence for those claims. This could involve:
>
> * Testing whether the system performs as claimed in practice, using a sample from the target group (i.e., as done in [3] with crowdworkers).
> * Citing a paper that does involve such an analysis.
> * Relying on the expressed needs of people in those groups, e.g., [4], [5].
>
> [3] Generating Fact Checking Briefs. Fan et al., 2020.
>
> [4] Automated Fact-Checking for Assisting Human Fact-Checkers. Nakov et al., 2021.
>
> [5] The Challenges of Online Fact Checking. Arnold, 2020.
>
> _(15) "Social Media Moderators (0%)" ??_
>
> We assume this question refers to lines 1269-1272 in the appendix, where we go through all the possible data actors. The list in the appendix includes our initial guesses as to what categories might be useful (see discussion of our annotation process on lines 146-148). We expected social media moderators to appear somewhere as data actors, but it didn't happen in any of the 100 papers we analysed. This is indeed quite surprising, especially as similar systems are in practice used for content moderation. This absence highlights why considering our recommendation to make the data actors explicit is important (as some authors may have implicitly considered them to be data actors).
>
> **Style and Presentation**
>
> Thanks for the pointers, very helpful!

---

### Official Review · Reviewer_21ex · 2023-08-04

**Soundness:** 1

**Excitement:**

2: Mediocre: This paper makes marginal contributions (vs non-contemporaneous work), so I would rather not see it in the conference.

**Paper Topic And Main Contributions:**

The paper is an analysis of 100 "highly cited" papers on the topic of automated fact-finding regarding the questions of (i) the motivation for the paper? Why was the solution being discussed? (ii) the means or mechanisms for doing this; the 'how', and (iii) who this proposed solution is intended for -- the participants in the scenarios being looked at.

The analysis was conducted by two annotators (presumably the authors) manually going through the abstracts and introductory sections of these 100 papers and marking passages/text-spans with a series of tags based on a set of guidelines that the paper proposes. The paper presents some of the statistics from this analysis.



**Questions For The Authors:**


- Are there scoring frameworks that could be designed to assign a numerical value to find potential candidates for fact-checking?

- What are good datasets for training and testing that might be assembled, either real or synthetic? (Generating synthetic datasets should be an interesting problem since the proposed framework could be inverted) Could some of these be done without doing deep NLP? (for instance, by doing purely propositional theorem proving, or by solving for evidence graphs, etc)



**Reasons To Accept:**

This is a discussion of a proposed structure for analyzing fact-checker algorithms, along with an overview of how this framework may be used to understand some of the more highly cited recent papers proposing solutions; as such, this is both interesting and important. Especially as we enter a time of increasingly easy generation of realistic looking narratives using AI, fact-checking, whether automated or not, will become increasingly critical. Having a consistent framework and baseline will make it easier to compare these systems against one another. As a starting point, future work in this area can extend the framework in interesting ways, stimulate discussions on how some of these metrics may be automated and how to compare or weight one dimension over another.



**Reasons To Reject:**

I was hoping that this would be an automated scoring algorithm for fact-checkers -- taking as input algorithms and datasets, and outputting a completeness and correctness score for each (essentially an equivalent of Precision and Recall in information retrieval). Perhaps even a machine learning based analysis of language and communication cues that may, in conjunction with background information, may be used to find potential candidates for further checking.

Instead, this is more of a thoughtful discussion or guidelines for how reviewers (or professional fact-checkers) might want to think about reading papers while evaluating them for their claims on fact-checking. The analysis of the 100 papers that the paper presents is somewhat underspecified, leading to ambiguity: for instance, while the authors point out that evidence is often lacking for claims being made in narratives (and missing entirely for 'vague narratives'), they also acknowledge that even though citations are included, they need not always be relevant. Checking a fact, even if there is a citation, requires chasing down the references, understanding them, and making a determination about the validity of the current 'fact'. The paper points out that citations/evidence in itself is not a silver bullet, since there are a number of ways in which the citations or evidence presented can be cause problems for the reader.

The paper is an interesting and very timely reminder of the problems that we should be thinking about, but EMNLP is not (in my view) the best forum for this work, given the focus of this work. An ethics or social science conference would be a better venue.

**Reproducibility:**

4: Could mostly reproduce the results, but there may be some variation because of sample variance or minor variations in their interpretation of the protocol or method.

**Reviewer Confidence:**

4: Quite sure. I tried to check the important points carefully. It's unlikely, though conceivable, that I missed something that should affect my ratings.

---

> ### Author Rebuttal · Authors · 2023-08-28
>
> We thank the reviewer for their comments. We are happy they found our paper interesting and timely, despite the low score.
>
> **Appropriateness for EMNLP:**
>
> We disagree that this paper is inappropriate for EMNLP. The conference explicitly invites surveys and position papers, as stated in the call for papers https://2023.emnlp.org/calls/main_conference_papers/. Moreover, this paper is a submission to the Ethics in NLP track, and EMNLP in general welcomes papers on ethics.
>
> We further want to clarify that our paper and guidelines are primarily intended for *authors* of automated fact-checking papers and related tasks (e.g., rumour and deception detection), although of course, we believe that readers and reviewers are likely to benefit as well. Many of the papers we analyse were published at EMNLP, including some of the most cited (for example, [1]) and several of the papers we analyse were produced in connection to the FEVER workshop which took place three times at EMNLP.
>
> [1] Truth of Varying Shades: Analyzing Language in Fake News and Political Fact-Checking. Rashkin et al., 2017.
>
> **Automating Annotation:**
>
> If we understand the reviewer correctly when they ask for *“an automated scoring algorithm for fact-checkers -- taking as input algorithms and datasets, and outputting a completeness and correctness score for each”*, the question is whether our content analysis process for papers could be automated. We believe automated content analysis is a very interesting direction. We would like to see how our analysis is received by the community first, to judge the usefulness of datasets and models for this; if the reception is positive, we would be happy to work towards this. The annotations we conducted will be useful to anyone who wishes to develop this kind of automated analysis.
>
> **Vague Narratives & Citations:**
>
> The reviewer highlights that *“while the authors point out that evidence is often lacking for claims being made in narratives (and missing entirely for 'vague narratives'), they also acknowledge that even though citations are included, they need not always be relevant.”* What we mean in our paper is that empirical evidence for claims in papers is lacking for two reasons: 1) because no citations are mentioned, or 2) sometimes when citations ARE mentioned, they do not support the argument being made (i.e., they are “irrelevant” to the claim).
>
> **Claim Detection Scoring Frameworks:**
>
> Systems for finding potential claims to fact-check are an active area of research. Claim detection has also been the subject of several shared tasks, including RumourEval and FACTIFY. These are included in our analysis, but we will deepen the discussion of these in the camera-ready if the paper is accepted.

---

### Meta-Review · Area_Chair_NYuZ · 2023-09-17

**Recommendation:** 5

**Metareview:**

The decision based on the reviews suggests acceptance, but with revisions needed. The reviewers have recognized the paper's systematic research approach, clear argumentation, and its contribution towards the understanding of automated fact-checking. They find the topic relevant and timely, and appreciate the detailed annotation guidelines and analysis procedure which lend credibility to the arguments in the paper.
However, they also identify certain limitations such as the paper's complexity and excessive use of specialized terminology which makes it difficult to understand. There was also criticism about the missing information on the papers examined. Therefore, while the paper has potential, it requires some revisions before publication.

---

### Decision · Program_Chairs · 2023-10-07

**Decision:**

Accept-Findings

**Comment:**

The decision based on the reviews suggests acceptance, but with revisions needed. The reviewers have recognized the paper's systematic research approach, clear argumentation, and its contribution towards the understanding of automated fact-checking. They find the topic relevant and timely, and appreciate the detailed annotation guidelines and analysis procedure which lend credibility to the arguments in the paper.
However, they also identify certain limitations such as the paper's complexity and excessive use of specialized terminology which makes it difficult to understand. There was also criticism about the missing information on the papers examined. Therefore, while the paper has potential, it requires some revisions before publication.